# Progestin-primed ovarian stimulation for fertility preservation in women with cancer: A comparative study

Francesca Filippi[1]*, Marco Reschini[1], Elisa Polledri[2,3], Anna Cecchele[1], Cristina Guarneri[1], Paola Vigano[1], Silvia Fustinoni[2,3], Peter Platteau[4], Edgardo Somigliana[1,2]

1 Infertility Unit, Fondazione IRCCS Ca' Granda Ospedale Maggiore Policlinico, Milan, Italy, 2 Department of Clinical Sciences and Community Health, Università Degli Studi di Milano, Milan, Italy, 3 Environmental and Industrial Toxicology Unit, Fondazione IRCCS Ca' Granda Ospedale Maggiore Policlinico, Milan, Italy, 4 Centre for Reproductive Medicine, Universitair Ziekenhuis Brussel, Vrije Universiteit Brussel, Brussels, Belgium

* francesca.filippi@policlinico.mi.it

## Abstract

### Background

In women scheduled for cancer treatment, oocytes cryopreservation is a well-established procedure. Random start protocols have been a substantial improvement in this setting, allowing to prevent delay in the initiation of cancer treatments. However, there is still the need to optimize the regimen of ovarian stimulation, to make treatments more patient-friendly and to reduce costs.

### Methods

This retrospective study compares two periods (2019 and 2020), corresponding to two different ovarian stimulation regimens. In 2019, women were treated with corifollitropin, recombinant FSH and GnRH antagonists. Ovulation was triggered with GnRH agonists. In 2020, the policy changed, and women were treated with a progestin-primed ovarian stimulation (PPOS) protocol with human menopausal gonadotropin (hMG) and dual trigger (GnRH agonist and low dose hCG) Continuous data are reported as median [Interquartile Range]. To overcome expected changes in baseline characteristics of the women, the primary outcome was the ratio between the number of mature oocytes retrieved and serum anti-mullerian hormone (AMH) in ng/ml.

### Results

Overall, 124 women were selected, 46 in 2019 and 78 in 2020. The ratio between the number of mature oocytes retrieved and serum AMH in the first and second period was 4.0 [2.3–7.1] and 4.0 [2.7–6.8], respectively (p = 0.80). The number of scans was 3 [3–4] and 3 [2–3], respectively (p<0.001). The total costs of the drugs used for ovarian stimulation were 940 € [774–1,096 €] and 520 € [434–564 €], respectively (p<0.001).

**Data Availability Statement:** All relevant data are within the manuscript and its Supporting information files.

**Funding:** This study was partially funded by Italian Ministry of Health - Current research IRCCS. The funds exclusively consisted in a financial support. No additional external funding was received for this study. The funders had no role in study design, data collection and analysis, decision to publish or preparation of the manuscript.

**Competing interests:** The authors of this manuscript have the following competing interests: E.S. received honoraria for presentations at meetings from Gedeon-Richter and Merck-Serono. He is also handling two grants of research from Ferring. None of the remaining authors reported any conflict of interest. This does not alter our adherence to PLOS ONE policies on sharing data and materials.

## Conclusions

Random start PPOS with hMG and dual trigger represents an easy and affordable ovarian stimulation protocol for fertility preservation in women with cancer, showing similar efficacy and being more friendly and economical.

## Introduction

During the last decade, the improved effectiveness of cancer cures has enhanced the attention on the quality of life of survivors, including the importance of preserving their capacity to have children. Infertility in cancer survivors is an important issue and could be a source of devastating psychological sufferance [1, 2]. Accordingly, fertility sparing treatments and technique of fertility preservation have spread. The most relevant Scientific Societies of oncology and reproductive medicine have provided specific guidelines [3–6]. Noteworthy, even women with early endometrial cancer are now considered for conservative management. They are treated hormonally to achieve transient cancer regression, thus allowing pregnancy seeking and postponing hysterectomy after delivery [7–10].

Oocytes cryopreservation plays a crucial role in preserving fertility capacity in young adult women. with malignancies [3–6]. In this setting, enhancing the number of collected oocytes, avoiding a delay in the initiation of oncological treatments, and limiting risks are essential aspects. This can improve effectiveness and safety. Substantial advancements have been obtained with the diffusion of random start protocols (i.e., promptly initiating the ovarian hyperstimulation regardless of the menstrual phase), dual stimulation (performing two hyperstimulation cycles one after the other), the use of aromatase inhibitors in women with hormone-sensitive cancers (to lower peripheral levels of potentially detrimental hormones) and ovulation triggering with Gonadotropin Releasing Hormone (GnRH) agonists rather than human Chorionic Gonadotropin (hCG) (to prevent Ovarian Hyperstimulation syndrome—OHSS) [11–14]. However, there is room for further improvements. There is the need to optimize the regimen of ovarian stimulation, to make treatments more patient-friendly and to reduce costs [15].

Following a local audit at the end of 2019 among physicians engaged in fertility preservation in our Unit, a new policy of ovarian stimulation for women with cancer was decided and implemented from the beginning of 2020. To reduce costs, we replaced GnRH antagonists with a progestin (progestin-primed ovarian stimulation—PPOS) protocol and substituted corrifollitropin and recombinant FSH with human menopausal gonadotropin (hMG). In addition, we introduced the dual trigger, associating low dose hCG to GnRH agonists to improve oocytes yield [16, 17]. In the present 'before and after' study (we compared two study periods of one year, before and after January 01st 2020), we evaluated ovarian stimulation outcomes in 2019 and in 2020 to draw some inferences on the effectiveness of the new regimen.

## Materials and methods

### Study design and participants

This 'before and after' study included all consecutive women with a first diagnosis of cancer who underwent ovarian hyperstimulation for oocytes cryopreservation at the Infertility Unit of the Fondazione IRCCS Ca' Granda Ospedale Maggiore Policlinico of Milan, Italy between January 2019, and December 2020. Two groups were compared, chronologically divided based on the date of implementation of the new policy of ovarian hyperstimulation (January 1st,

2020). During the same study period (2019–2020), our unit was also engaged in a parallel study aimed at evaluating follicular fluid steroids concentrations in women with cancer undergoing random start ovarian stimulation protocols. This situation offered us the additional opportunity to obtain information on the local levels of steroids hormones detected for the two different regimens under study.

### Ethical considerations

The study was approved by the local Ethical Committee (Milano area B). Women signed an informed consent to participate to the study on follicular fluid steroids concentrations. Conversely, since the 'before and after' part of the study was decided *a posteriori*, they did not sign a specific informed consent for the comparisons of clinical outcomes. However, all women undergoing fertility preservation in our Unit routinely signed an informed consent for their anonymized data to be used for research purposes.

### Protocols of treatment in the two study periods

During the whole study period, ovarian hyperstimulation was performed according to a random start approach. Gonadotropins were initiated on the day of referral regardless of the phase of the menstrual cycle. In hormone-sensitive cancers, in all the studied period, women were also given oral letrozole 5 mg (Letrozolo®, TEVA, Italy) daily for the whole duration of ovarian hyperstimulation and up to 5–7 days after oocytes retrieval. The dose of gonadotropins was decided based on the expected ovarian response, i.e., considering clinical data (age, menstrual cycle duration), antral follicular count (AFC) assessed with an ultrasound scan at first referral and serum anti-mullerian hormone (AMH). Women were monitored with serial transvaginal ultrasounds (US) and peripheral hormonal dosages if needed [15]. Serum AMH concentration was tested in all women prior to initiate the ovarian hyperstimulation. The commercially available Beckman Coulter Gen II ELISA assay on the automated GEMINI platform (STRATEC Biomedical AG, Germany) after dilution with assay buffer was used.

In the first study period (January to December 2019), women initiated ovarian hyperstimulation with the long-acting corifollitropin alfa (Elonva®, Merck Sharp & Dohme, UK) 100 or 150 mcg s.c. according to body weight (< or ≥ 60 kg). Five-seven days later, recombinant FSH daily s.c. (150/225 IU) (Gonal-F®, Merck-Serono, Italy) was administered, if needed. Pituitary suppression was obtained with GnRH antagonist s.c. (Fyremadel® 0.25 mg Ferring, Denmark) administered daily, initiating when the leading follicle reached a mean diameter of 13–14 mm, and continued up to the time of ovulation trigger. Final oocyte maturation was triggered with GnRH agonists (Fertipeptyl® 0.2 mg, Ferring, Denmark) when at least three follicles had a mean diameter of 18 mm (Fig 1).

Starting from January 2020, ovarian hyperstimulation was obtained using hMG s.c. (Meropur®, Ferring, Denmark) 150/225 IU, daily. Contemporary, medroxy-progesterone acetate (MPA) tablets 10 mg (Farlutal, Pfeizer, US) were administered daily until the day of trigger. The dose of MPA was chosen according to previous publications to ensure pituitary inhibition [18]. Final oocyte maturation was triggered with a combination of GnRH agonists (Fertipeptyl® 0.2 mg, Ferring, Denmark) and Chorionic gonadotrophin (hCG) 1,000 IU (Gonasi HP, Ibsa, Switzerland) when at least three follicles had a mean diameter of 18 mm (Fig 1).

### Follicular fluid analyses

During the whole study period (2019–2020), oocytes were collected 36 hours after triggering through transvaginal ultrasound-guided oocyte retrieval. The protocols of oocytes aspiration, oocytes handling and cryostorage were standardized and reported elsewhere in detail [19, 20].

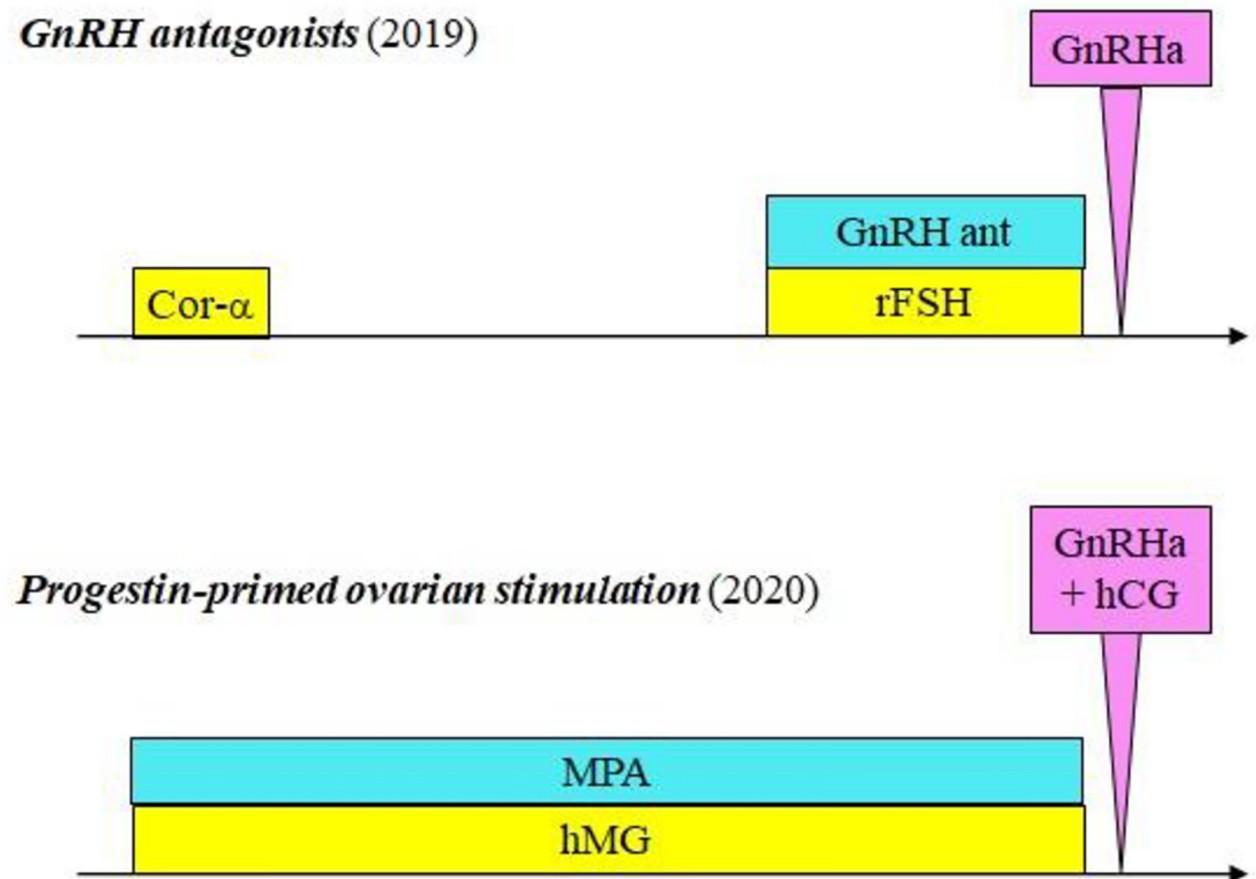

**Fig 1. The figure illustrates the two compared ovarian hyperstimulation regimens.** In the *upper part of the figure*, the regimen used in 2019. Women received corifollitropin-alfa and 6–7 days later recombinant FSH daily, if necessary. Since the observation of a leading follicle with a mean diameter above 13–14 mm, they were prescribed also GnRH antagonist daily. The final maturation was obtained with GnRH agonist 0.2 mg. In the *lower part of the figure*, the regimen used in 2020 (progestin-primed ovarian stimulation). Women received both hMG and oral MPA daily. Final oocytes maturation was obtained with the concomitant administration of GnRH agonist 0.2 mg, plus hCG 1,000 IU. GnRH: Gonadotropin Releasing hormone. Cor-α: Corifollitropin-alfa. GnRH ant: GnRH antagonist. GnRHa: GnRH agonist. rFSH: recombinant FSH. MPA: medroxy-progesterone acetate. hMG: human Menopausal Gonadotropin. hCG: human Choriogonadotropin.

The methodology of follicular fluids collection and steroids assessment is also reported in detail elsewhere [21]. Briefly, the pool of follicular fluids was collected, and then centrifuged and cryopreserved at -20˚C. The samples were concomitantly thawed and levels of 15 steroids were then assayed by liquid chromatography tandem mass spectrometry. The following 15 steroids were tested: 11-deoxycorticosterone, 11-deoxycortisol, 17-OH-progesterone, 21-deoxycortisol, aldosterone, androstenedione, corticosterone, cortisol, cortisone, dehydroepiandrosterone (DHEA), dehydroepiandrosterone sulfate (DHEAS), dihydrotestosterone (DHT), estradiol, progesterone, testosterone. Results were presented separately for women who did and did not receive letrozole because of the well-known relevant effects of this drug on steroidogenesis [21].

### Studied variables

Clinical information regarding patients' history, cancer diagnosis, US monitoring and oocytes retrieved was obtained from patients' charts. The costs of the drugs were obtained as described

in detail elsewhere [22]. Gonadotropins were fully refunded in Lombardy. GnRH agonists and antagonists, hCG and MPA were conversely paid out of pocket by the patients. For the analyses, we did not distinguish between those that were or were not refunded and considered them globally. Only drugs used for ovarian hyperstimulation were included (i.e., gonadotropins, GnRH antagonists, GnRH agonists, hCG, and MPA). The costs of the drugs were obtained referring to the local Italian Agency of drugs—Agenzia Italiana del farmaco (AIFA) (*Farmadati sftw access 02/22*).

## Statistical analyses and study power

Data collected were transferred in Statistical Package for Social Science (SPSS 26.0, IL, USA) database for subsequent analyses. Differences between the two study groups were tested using Student's *t* test or Mann-Whitney test or Fisher Exact test, as appropriate. The primary outcome was the ratio between the number of mature oocytes retrieved and serum AMH. This outcome was chosen to protect the findings from possible differences in ovarian reserve between the two study periods. To further limit confounders, we set the whole duration of the study period to two years (one year per group). Given an expected number of about 60 cases per year, a mean ± SD number of oocytes stored (10 ± 6) and the setting of type I and II errors at 0.05 and 0.20 respectively, our sample size could allow detection of differences > 3 oocytes.

## Results

Overall, 124 women were selected, 46 cryopreserved oocytes in 2019 while 78 on the following year. Twelve women performed dual stimulation and only their first hyperstimulation cycle was accounted for the study purposes. Baseline characteristics of the two groups are shown in Table 1. Indications to oocytes cryopreservation were similar in the two study periods. However, a statistically significant difference emerged for serum AMH, levels being lower in the second period.

**Table 1. Baseline clinical characteristics of the study groups.**

| Characteristics | GnRH antagonists | PPOS | p |
|---|---|---|---|
| | n = 46 | n = 78 | |
| Age (years) | 32 [28–36] | 31 [27–35] | 0.21 |
| BMI (Kg/m$^2$) | 21.4 [19.6–23.2] | 21.5 [19.5–24.4] | 0.45 |
| Previous deliveries | 6 (13%) | 14 (18%) | 0.62 |
| Seeking pregnancy at the time of the diagnosis | 4 (9%) | 11 (14%) | 0.57 |
| Indication to oocytes cryopreservation | | | 0.39 |
| Breast cancer | 26 (57%) | 31 (40%) | |
| Hematological cancers | 10 (22%) | 22 (28%) | |
| Ovarian cancer | 3 (7%) | 4 (5%) | |
| Central nervous system | 1 (2%) | 3 (4%) | |
| Others cancer | 6 (13%) | 18 (23%) | |
| Duration of cycles (days) | 28 [28–30] | 28 [28–30] | 0.74 |
| Serum AMH (ng/ml) | 2.80 [1.41–4.46] | 1.88 [0.94–3.15] | 0.02 |
| Total AFC | 18 [12–27] | 15 [10–23] | 0.15 |

AFC: Antral Follicle Count. AMH: Anti-mullerian hormone

PPOS: Progestin Primed Ovarian Stimulation

Data are reported as mean ± SD or median [interquartile range] or number (percentage)

**Table 2. Baseline clinical characteristics of the study groups.**

| Characteristics | GnRH antagonists | PPOS | p |
|---|---|---|---|
| | n = 46 | n = 78 | |
| Cycle phase at initiation of hyper-stimulation | | | 1.00 |
| Follicular phase | 28 (61%) | 48 (62%) | |
| Luteal phase | 18 (39%) | 30 (38%) | |
| Letrozole use | 22 (48%) | 26 (33%) | 0.13 |
| N. of cancelled cycles | 3 (7%) | 5 (6%) | 1.00 |
| Total dose of FSH (IU) [a,b] | 2,025 [1,800–2,450] | 2,000 [1,800–2,400] | 0.95 |
| Duration of stimulation (days) [a] | 10 [8–10] | 10 [9–12] | 0.004 |
| N. of developed follicles (diameter ≥11 mm) [a] | 19 [12–24] | 15 [10–20] | 0.03 |
| N. of developed codominant follicles (diameter ≥15 mm) [a] | 10 [7–14] | 8 [6–12] | 0.08 |
| N. of oocytes retrieved [a] | 16 [10–21] | 10 [5–17] | 0.008 |
| N. of mature oocytes frozen [a] | 10 [6–18] | 9 [4–14] | 0.05 |

Data are reported as mean ± SD or median [interquartile range] or number (percentage)

PPOS: Progestin-primed ovarian stimulation

[a] Refers to women who completed the cycle (43 and 73 in the GnRH antagonist and PPOS groups, respectively)

[b] Corrifollitropin alfa 100 and 150 mcg were considered equivalent to 150 and 200 IU per day for 7 days, respectively.

In the first part, three women did not complete their fertility preservation cycle: two decided to anticipate the beginning of chemotherapy and one was stopped because of inadequate follicular growth. In the second part, five women dropped out: one due to personal choice and four for inadequate follicular growth. Therefore, a total of 43 and 73 cycles were available for oocytes retrieval outcomes in 2019 and 2020, respectively. Cycle outcome of the two study periods is illustrated in Table 2. In the first period, the median [IQR] number of GnRH antagonist injections was 4 [3–5]. In both study periods, we did not observe spontaneous ovulations during the stimulation. Some variables significantly differed. The number of oocytes retrieved and the number of mature oocytes frozen were significantly lower in the second period. However, the ratio between number of mature oocytes and AMH did not differ (Fig 2). Specifically, the median [IQR] rate of retrieved oocytes adjusted per AMH in the first and second period was 5.2 [3.3–8.1] and 5.5 [3.6–8.6], respectively (p = 0.76). The median [IQR] rate of mature oocytes adjusted per AMH in the first and second period was 4.0 [2.3–7.1] and 4.0 [2.7–6.8], respectively (p = 0.80).

The number of ultrasounds was significantly lower in women treated in the second period. The median [IQR] number of scans was 3 [3–4] and 3 [2–3] in the first and second period, respectively (p<0.001). The median [IQR] number of injections (including those needed for gonadotropins, GnRH antagonists, GnRH agonists, and hCG) was 12 [10–14] and 13 [12–15] in the first and second period, respectively (p = 0.01). The total costs of the drugs used for ovarian hyperstimulation (including those needed for gonadotropins, GnRH antagonists, GnRH agonists, hCG, and MPA) in the two study periods were 940 € [IQR: 774–1,096 €] and 520 € [IQR: 434–564 €], respectively (p<0.001).

Table 3 shows sex steroids levels in the two study groups, considering separately those who did and did not receive letrozole. Some statistically significant differences emerged. For women who did not receive letrozole, PPOS-treated subjects (second period) had higher levels of Aldosterone, 11-DeoxyCortisol, Androstenedione, Testosterone and lower levels of Cortisol, Corticosterone, DHEAS, 17-OH-Progesterone. For those who received letrozole, PPOS-treated subjects had higher levels of Aldosterone, 11-DeoxyCortisol, testosterone and lower levels of 17-OH-Progesterone.

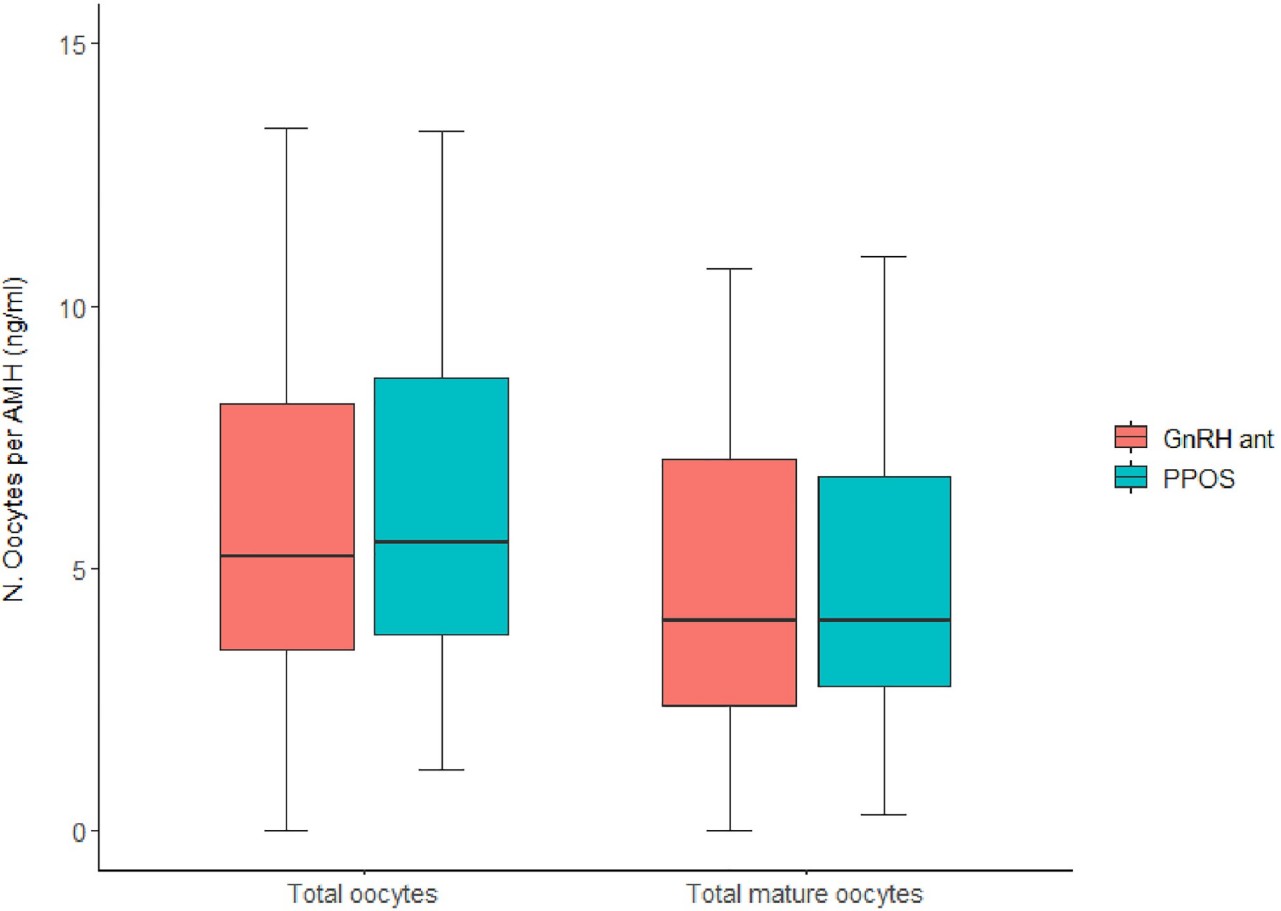

**Fig 2. Ratio of oocytes retrieved per serum AMH (reported in ng/ml).** The left part of the figure refers to the total number of oocytes retrieved. The right part refers to the total number of mature oocytes retrieved and thus cryopreserved. The first study period (simplified in the figure with GnRH ant, to underline the use of GnRH antagonists) is depicted in red. The second study period (simplified in the figure with PPOS, corresponding to progesterone primed ovarian stimulation) is depicted in blue. No statistically significant difference emerged for both comparisons (p = 0.76 and p = 0.80, respectively).

## Discussion

### Rationale of the study

The ability of progesterone administered from the early follicular phase to inhibit follicular growth and LH surge by blocking the estradiol signal and slowing the LH pulse frequency, thus decreasing its plasma content, is known for a long time [23]. Since 2015, when PPOS protocol was firstly proposed by Kuang and coworkers [18], several studies have applied this regimen to patients with different characteristics. A systematic review and meta-analysis concluded that there is high-quality evidence supporting the effectiveness of PPOS in retrieving a similar number of oocytes without affecting the duration of stimulation and gonadotropin consumption [24]. There is also evidence that oocytes competence is not affected. The rate of euploid blastocysts was shown to be similar [25]. Accordingly, another meta-analysis failed to show any difference in live birth rate between PPOS and GnRH antagonist protocols [26]. Finally, Zolfaroli et al. recently published a meta-analysis on 9,274 newborn babies and did not show significant differences in the risk of congenital malformations and low birth weight between women treated with PPOS protocol and those treated with GnRH analogues [27].

**Table 3. follicular fluid steroids concentration in the two study periods, presented separately according to the concomitant assumption of letrozole.**

| Steroids | Without letrozole | | | With letrozole | | |
|---|---|---|---|---|---|---|
| | GnRH antagonists | PPOS | p | GnRH antagonists | PPOS | p |
| | n = 23 | n = 48 | | n = 18 | n = 24 | |
| Aldosterone (µg/l) | 0.06 [0.03–0.13] | 0.10 [0.06–0.20] | 0.03 | 0.05 [0.01–0.09] | 0.10 [0.08–0.14] | 0.002 |
| Cortisol (µg/l) | 70.5 [52.1–82.3] | 52.6 [30.0–77.3] | 0.03 | 50.4 [39,2–54,0] | 52.0 [39.9–63.1] | 0.61 |
| Cortisone (µg/l) | 13.2 [10.6–20.4] | 14.2 [8.0–18.9] | 0.43 | 13.3 [9.9–16.8] | 14.3 [12.6–16.5] | 0.48 |
| 11-Deoxy-Cortisol (µg/l) | 0.94 [0.72–1.63] | 1.60 [1.25–2.34] | 0.004 | 1.25 [1.03–2.64] | 2.40 [1.62–3.26] | 0.02 |
| 21-Deoxy-Cortisol (µg/l) | 0.17 [0.08–0.25] | 0.17 [0.10–0.39] | 0.47 | 0.18 [0.11–0.35] | 0.16 [0.08–0.29] | 0.58 |
| Corticosterone (µg/l) | 3.35 [2.78–4.33] | 2.63 [1.36–3.68] | 0.02 | 2.68 [2.11–3.93] | 2.31 [1.74–3.80] | 0.35 |
| 11-Deoxy-Corticosterone (µg/l) | 29.0 [22.8–35.2] | 26.8 [21.7–36.0] | 0.43 | 21.7 [20.3–27.9] | 23.7 [20.0–30.7] | 0.65 |
| DHEAS (µg/l) | 1,442 [1,226–2,048] | 1,188 [758–1,662] | 0.01 | 1,708 [997–2,122] | 1,446 [901–2,219] | 0.63 |
| DHEA (µg/l) | 6.49 [4.48–9.91] | 6.58 [4.24–11.57] | 0.98 | 8.70 [6.85–12.30] | 7.61 [5.25–9.82] | 0.24 |
| Estradiol (µg/l) | 286 [188–498] | 418 [303–555] | 0.06 | 110 [50–217] | 121 [83–178] | 0.72 |
| Androstenedione (µg/l) | 4.67 [1.90–11.21] | 15.0 [4.7–42.4] | 0.02 | 182.6 [133.6–266.8] | 185.7 [147.0–225.5] | 0.96 |
| Testosterone (µg/l) | 0.15 [0.04–0.36] | 0.55 [0.17–2.36] | 0.002 | 23.5 [19.6–39.2] | 33.6 [30.9–53.7] | 0.01 |
| DHT (µg/l) | 0.07 [0.03–0.25] | 0.08 [0.03–0.21] | 0.79 | 0.26 [0.14–0.35] | 0.36 [0.17–0.58] | 0.27 |
| 17-OH-Progesterone (µg/l) | 469 [374–677] | 361 [299–447] | 0.008 | 642 [528–759] | 443 [346–730] | 0.01 |
| Progesterone (µg/l) | 566 [464–679] | 524 [416–648] | 0.22 | 581 [515–640] | 517 [490–765] | 0.20 |

However, even if there is evidence supporting the effectiveness of PPOS for egg freezing in non-oncological contexts [24, 28], specific data in the subgroup of women with cancer is very scant.

## Main findings and what it adds to what we already know

Our study aimed at evaluating the effectiveness of a novel approach of ovarian stimulation for fertility preservation in women with cancer, an approach combining random start, PPOS with hMG, and dual trigger. To this aim, we compared two study periods of one year, before and after the implementation of the new regimen. The new approach emerged to be a valid option. We observed a lower number of mature oocytes retrieved but this difference completely disappeared when adjusting for basal AMH. In addition, even if the duration of ovarian hyperstimulation resulted one day longer and women required one more injection, the total costs of the drugs used for ovarian stimulation was lower, and women required less US scans for monitoring follicular growth. Overall, the new regimen is less expensive and may also be more friendly for women.

Results on follicular fluid composition of steroids hormones in women treated with POPP or GnRH antagonists is a corollary but interesting new finding of our study. Several differences emerged, suggesting that the PPOS protocols with hMG differently affect the steroid cascade compared to GnRH antagonists with recombinant FSH. Significant changes were documented for 17-OH-Progesterone, aldosterone, 11-DeoxyCortisol, corticosterone and DHEAS. These differences could be observed in both women who did and did not receive letrozole. Fig 3 illustrates the cascade to facilitate interpretation.

The difference in 17-OH-Progesterone was the most expected, being the direct conversion of progesterone. The exogenous administration of a progestin may interfere with local production. Other compounds could be also derived from this disarranged equilibrium being subsequent steps of the arms arising from progesterone. The increase in androgens (testosterone and androstenedione) could be due to the use of hMG rather than recombinant FSH as well as

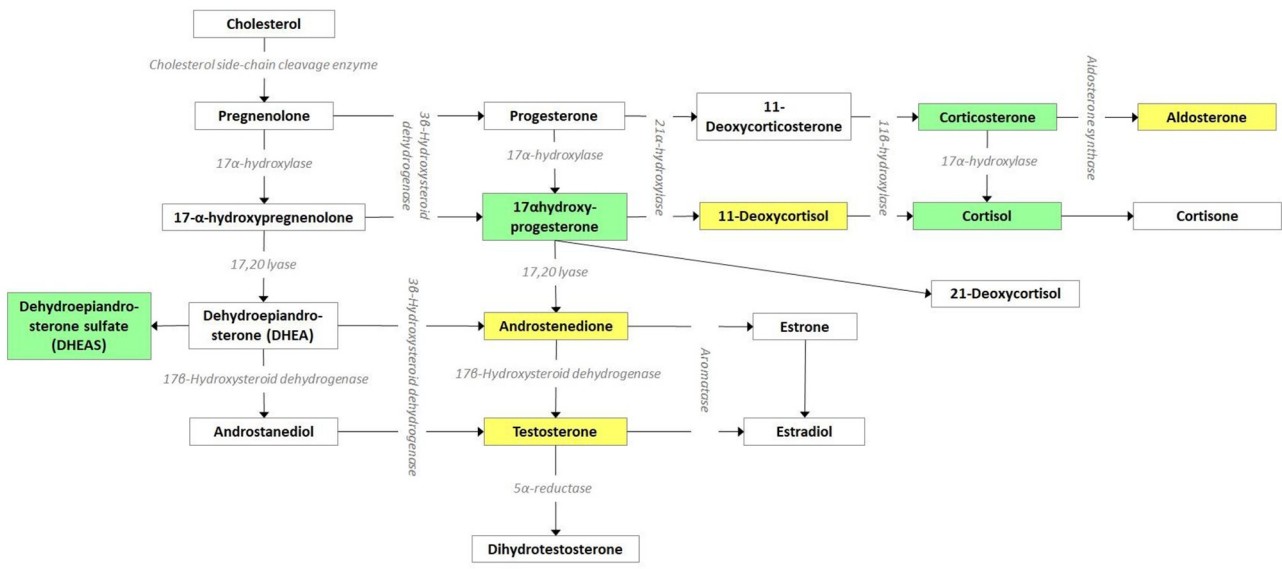

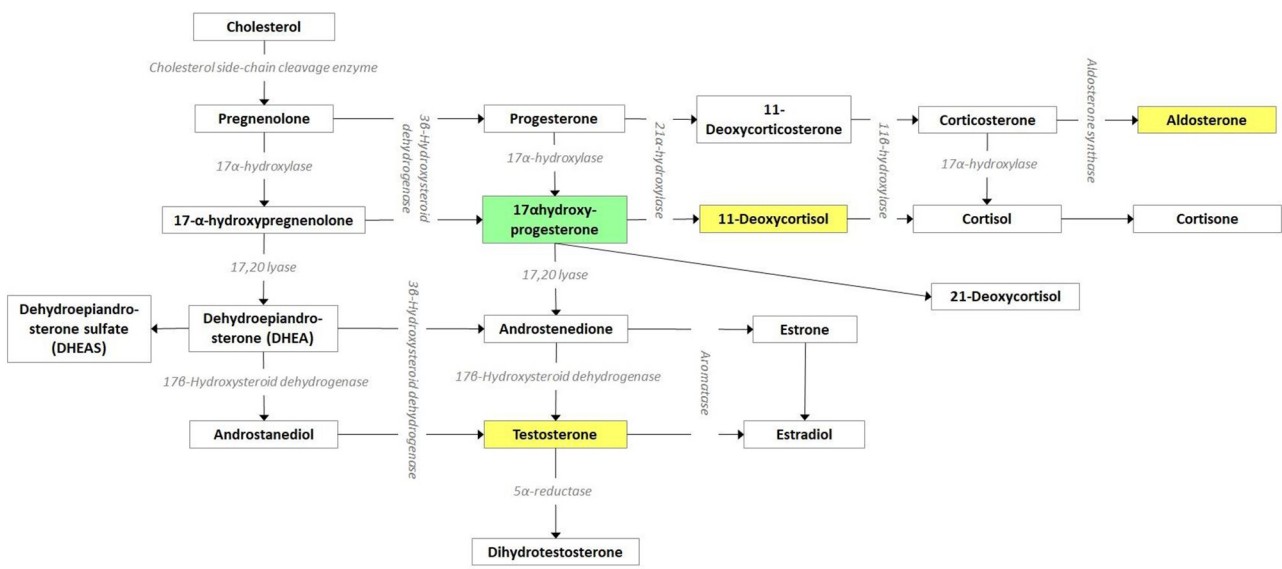

**Fig 3. The follicular fluid steroids cascade in the two study groups.** Data are presented separately for those who did (lower panel) and did not (higher panel) receive letrozole. Statistically significant increases were highlighted in yellow. Statistically significant decreases are highlighted in green (for details, see Table 3).

the addition of low dose of hCG for the trigger. Difference observed for DHEA levels in the subgroup of women who did not receive letrozole is more difficult to interpret and could be attributed to a type I error. We may also interpret the differences observed in patients with and without letrozole as a type II error because of the smaller sample size of the latter group. The four compounds who were found to differ in the letrozole group were also shown to differ

in the non-letrozole group. In addition, for the compounds that differed only in the non-letrozole group, non-significant trends in the same direction were observed in the letrozole group. In general, however, these statistically significant differences may not turn into negative biological consequences for the oocytes.

## Subject of the discussion

As previously mentioned, the most important outcome is represented by the oocytes quality and related chances to lead to live births. Our study cannot provide evidence on this point. Data onn follicular fluid composition is indirect. To note, the observed differences in sex steroids follicular concentration may have an endocrinological impact (growing follicles produce hormones aimed at regulating functions and modulating feedbacks in distant organs) but may be irrelevant to oocytes quality.

## Comparison with previous studies, and possible agreement and disagreement

To our knowledge, this is the largest contribution on PPOS in women with cancer. We identified only one previous study on this topic. Huang et al. recently published a review of their five years activity of fertility preservation in women with cancer, comparing the outcome of 30 PPOS and 56 GnRH antagonists' cycles. No differences in the number of retrieved oocytes emerged [29]. The retrospective study design (in particular the allocation based on physicians' decision) and the long study period (five years) inevitably exposed these findings to important confounders. In this context, our 'before and after' study design should be considered more reliable. The change in the prescribed regimen occurred at a precise time point and no other policy changes occurred. Even if this type of study design cannot fully overcome all possible confounders, it has some important advantages [15, 30].

## Strengths and limitations

The non-randomized design of our study is an important limitation that must be acknowledged. Results are exposed to possible confounders. Of relevance here is that basal characteristics of the two study groups differed. A higher number of women were treated in 2020, mainly due to an increase in the number of hematological cases. Most importantly, serum AMH was significantly lower in the second period and a trend for lower AFC also emerged (p = 0.08). For this reason, the main outcome, i.e., the number of mature oocytes retrieved was adjusted for AMH. However, we cannot rule out other confounders and results cannot therefore be considered definitive. On the other hand, it is difficult to conduct randomized controlled trials (RCTs) in the clinical setting of oocytes cryopreservation for cancer [15]. Patients are distressed by the recent diagnosis and ovarian stimulation must be promptly started to avoid delays in the initiation of oncological treatments. The time needed for a well-informed and shared involvement in a RCT is lacking. Moreover, the clinical relevance of the issue is too modest to justify additional stress to the patients. As alluded above, even if evidence on PPOS in women with cancer is scant [29], there is a growing and convincing literature supporting this approach in other contexts such as oocytes donation or Preimplantation Genetic Testing [24, 31].

Some additional limitations of our study should be acknowledged. The relatively small sample size, the retrospective design and the non-randomized allocation have been already discussed above. We herein believe important to mention the issue of oncological safety, a topic that deserves extreme attention. Of relevance here is the use of PPOS in women with hormone sensitive cancers, and in particular hormone sensitive breast cancer. One could speculate that the progestin may stimulate the growth and diffusion of the malignancy. It is unlikely that a

short exposure (10–15 days) to a low dose progestin may be of clinical relevance, but scientific data to rule out this concern is lacking. Further studies are required.

## Clinical implications

Compared to the subcutaneous injections needed for GnRH antagonist administration, the oral administration of progestins is more tolerable by the women. A close monitoring to identify the time of GnRH antagonist initiation (leading follicle with a mean diameter >13–14 mm) is also not needed. Our results are in line with recent findings suggesting that PPOS could be more cost-effective for planned freeze all cycles such as oocytes donation or fertility preservation [32]. D'Argent et al also showed that PPOS protocols could be more cost-effective compared with GnRH antagonists when used for fertility preservation in women with endometriosis [33]. Our study did not specifically investigate cost-effectiveness, but we can infer similar results. Of interest here is the observation that the number of transvaginal ultrasound scans to monitor follicular growth was inferior in PPOS, thus reducing indirect costs such as the loss of working hours. To note, cancer patients are often required several medical exams before the beginning of oncological treatments and limiting the number of ultrasound scans can reduce their clinical burden and psychological stress, increasing their adherence to treatments. The opportunity to limit the commuting to hospital revealed to be useful also in the peculiar situation of COVID-19 pandemic.

## Conclusions

In conclusion, random start PPOS with hMG and dual trigger represents an easy and affordable ovarian stimulation protocol for fertility preservation in women with cancer, showing similar results to random start with GnRH antagonist plus recombinant FSH protocol but being potentially more friendly and cheap. Additional evidence is needed to rule out a clinical relevance of the detected changes in local steroids hormones and to demonstrate the safety for hormone sensitive cancers. More in general, future efforts are needed to identify the most suitable protocol to be used in this clinical setting and to better tailor treatments according to the clinical characteristics of the patients.

## Supporting information

**S1 Dataset.**
(XLSX)

## Author Contributions

**Conceptualization:** Francesca Filippi, Paola Vigano, Edgardo Somigliana.

**Data curation:** Francesca Filippi, Anna Cecchele, Cristina Guarneri, Paola Vigano, Silvia Fustinoni, Edgardo Somigliana.

**Formal analysis:** Marco Reschini, Elisa Polledri, Silvia Fustinoni, Edgardo Somigliana.

**Methodology:** Marco Reschini, Elisa Polledri, Edgardo Somigliana.

**Validation:** Cristina Guarneri, Edgardo Somigliana.

**Writing – original draft:** Francesca Filippi, Paola Vigano, Edgardo Somigliana.

**Writing – review & editing:** Francesca Filippi, Marco Reschini, Elisa Polledri, Anna Cecchele, Cristina Guarneri, Paola Vigano, Silvia Fustinoni, Peter Platteau.

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
