## [Decision Letter · Decision Letter 0]

3 Oct 2022

PONE-D-22-23405Optimizing ovarian hyperstimulation for fertility preservation in women with cancer a ‘before and after’ studyPLOS ONE

Dear Dr. Francesca Filippi,

Thank you for submitting your manuscript to PLOS ONE. After careful consideration, we feel that it has merit but does not fully meet PLOS ONE’s publication criteria as it currently stands. Therefore, we invite you to submit a revised version of the manuscript that addresses the points raised during the review process.

Please submit your revised manuscript by November 1, 2022. If you need more time than this to complete your revisions, please reply to this message or contact the journal office at plosone@plos.org. Please include the following items when submitting your revised manuscript:A rebuttal letter that responds to each point raised by the academic editor and reviewer(s). You should upload this letter as a separate file labeled 'Response to Reviewers'.A marked-up copy of your manuscript that highlights changes made to the original version. You should upload this as a separate file labeled 'Revised Manuscript with Track Changes'.An unmarked version of your revised paper without tracked changes. You should upload this as a separate file labeled 'Manuscript'.If applicable, we recommend that you deposit your laboratory protocols in protocols.io to enhance the reproducibility of your results. Protocols.io assigns your protocol its own identifier (DOI) so that it can be cited independently in the future. For instructions see: https://journals.plos.org/plosone/s/submission-guidelines#loc-laboratory-protocols. Additionally, PLOS ONE offers an option for publishing peer-reviewed Lab Protocol articles, which describe protocols hosted on protocols.io. Read more information on sharing protocols at https://plos.org/protocols?utm_medium=editorial-email&utm_source=authorletters&utm_campaign=protocols.

We look forward to receiving your revised manuscript.

Kind regards,

Gulzhanat Aimagambetova

Academic Editor

PLOS ONE

Journal Requirements:

"This study was partially funded by Italian Ministry of Health – Current research IRCCS."

"I have read the journal's policy and the authors of this manuscript have the following competing interests: E.S. received honoraria for presentations at meetings from Gedeon-Richter and Merck-Serono. He is also handling two grants of research from Ferring. None of the remaining authors reported any conflict of interest."

Reviewers' comments:

Reviewer's Responses to Questions

**Comments to the Author**

1. Is the manuscript technically sound, and do the data support the conclusions?

Reviewer #1: Partly

Reviewer #2: Yes

2. Has the statistical analysis been performed appropriately and rigorously? 

Reviewer #1: Yes

Reviewer #2: Yes

3. Have the authors made all data underlying the findings in their manuscript fully available?

Reviewer #1: Yes

Reviewer #2: Yes

4. Is the manuscript presented in an intelligible fashion and written in standard English?

Reviewer #1: Yes

Reviewer #2: Yes

5. Review Comments to the Author

Reviewer #1: The title of the manuscript sounds confusing. It will benefit if rephrased for more clarity.

The abstract provides some understanding of the study, however, should be improved to represent the study in a clearer way. In the methods section of the abstract, the study design should be explained. It is not clear what does "before and after '' mean?

The manuscript draft is landscape oriented. Is there any practical need for it? Please follow the PLOS ONE journal requirements at https://journals.plos.org/plosone/s/submission-guidelines.

The introduction part provides a clear rationale for the study. However, it is too narrow and the text will benefit if reworked. In particular, more updated references are suggested. For example for the sentences in lines 39-41 and 42-46 the following more updated sources could be used - doi: 10.3332/ecancer.2020.1030; doi: 10.3390/jcm11010196; doi: 10.3390/ijms222111825; doi: 10.1007/s00404-018-5006-z.

Moreover, information on the epidemiology of GYN cancers in general and among young women and the current situation in your country with the fertility preservation approach could improve the introduction part.

The methods section is detailed. However, poorly structured. Please provide the following items (better to highlight with the relevant subheadings, which will make the text more comprehensible for a potential audience): Study Design, study participants/subjects, Study instruments, variables, study protocol, ethical considerations, and statistical analyses.

The text in lines 107-114 of the methods section looks confusing. It might sound better if moved to the results section. However, it is up to the authors to decide.

The results are promising and presented clearly, supported by tables and figures.

The discussion part is interesting but will improve if restructured according to the suggested outline (the following items should be mentioned): Rationale of the study (why it was done); Main findings of the study; What makes your study unique; What it adds to what we already know; Subject of the discussion, Comparison with previous studies, agreement and disagreement with the studies compared; study strengths and limitations; Clinical implication.

Reviewer #2: I read with great interest the manuscript, which falls within the aim of this Journal. In my honest opinion, the topic is interesting enough to attract the readers’ attention. Nevertheless, the authors should clarify some points and improve the discussion, as suggested below.

Authors should consider the following recommendations:

- I recommend to discuss, at least briefly, novel pieces of evidence about fertility-preservation in case of early-stage endometrial cancer, since it is one of the most common gynecological malignancies in women of reproductive age (authors may refer to: PMID: 36143933).

- Authors may consider to add further details about the psychological impact of cancer in women wishing to preserve their fertility (PMID: 28275393).

6. PLOS authors have the option to publish the peer review history of their article (what does this mean?). If published, this will include your full peer review and any attached files.

Reviewer #1: No

Reviewer #2: No

---

## [Author Response · Author response to Decision Letter 0]

30 Nov 2022

REPLY: Done

"This study was partially funded by Italian Ministry of Health – Current research IRCCS."

REPLY: Done

"I have read the journal's policy and the authors of this manuscript have the following competing interests: E.S. received honoraria for presentations at meetings from Gedeon-Richter and Merck-Serono. He is also handling two grants of research from Ferring. None of the remaining authors reported any conflict of interest."

REPLY: Done

4. Please review your reference list to ensure that it is complete and correct. If you have cited papers that have been retracted, please include the rationale for doing so in the manuscript text or remove these references and replace them with relevant current references. Any changes to the reference list should be mentioned in the rebuttal letter that accompanies your revised manuscript. If you need to cite a retracted article, indicate the article’s retracted status in the References list and also include a citation and full reference for the retraction notice.

REPLY: Done

Reviewers' comments:

Reviewer's Responses to Questions

Comments to the Author

1. Is the manuscript technically sound, and do the data support the conclusions?

Reviewer #1: Partly

Reviewer #2: Yes

REPLY: We thank the reviewers for appreciating our manuscript and we agree on the general judgments (albeit not univocal).

2. Has the statistical analysis been performed appropriately and rigorously?

Reviewer #1: Yes

Reviewer #2: Yes

REPLY: Thank you again for the appreciation. 

3. Have the authors made all data underlying the findings in their manuscript fully available?

The PLOS Data policy requires authors to make all data underlying the findings described in their manuscript fully available without restriction, with rare exception (please refer to the Data Availability Statement in the manuscript PDF file). The data should be provided as part of the manuscript or its supporting information or deposited to a public repository. For example, in addition to summary statistics, the data points behind means, medians and variance measures should be available. If there are restrictions on publicly sharing data—e.g., participant privacy or use of data from a third party—those must be specified.

Reviewer #1: Yes

Reviewer #2: Yes

REPLY: Thank you again

---

## [Decision Letter · Decision Letter 1]

26 Dec 2022

Progestin-primed ovarian stimulation for fertility preservation in women with cancer: a comparative study

PONE-D-22-23405R1

Dear Dr. Francesca Filippi,

We’re pleased to inform you that your manuscript has been judged scientifically suitable for publication and will be formally accepted for publication once it meets all outstanding technical requirements.

Kind regards,

Gulzhanat Aimagambetova

Academic Editor

PLOS ONE

Reviewers' comments:

Reviewer's Responses to Questions

**Comments to the Author**

1. If the authors have adequately addressed your comments raised in a previous round of review and you feel that this manuscript is now acceptable for publication, you may indicate that here to bypass the “Comments to the Author” section, enter your conflict of interest statement in the “Confidential to Editor” section, and submit your "Accept" recommendation.

Reviewer #1: All comments have been addressed

Reviewer #2: All comments have been addressed

2. Is the manuscript technically sound, and do the data support the conclusions?

Reviewer #1: Yes

Reviewer #2: Yes

3. Has the statistical analysis been performed appropriately and rigorously? 

Reviewer #1: Yes

Reviewer #2: Yes

4. Have the authors made all data underlying the findings in their manuscript fully available?

Reviewer #1: Yes

Reviewer #2: Yes

5. Is the manuscript presented in an intelligible fashion and written in standard English?

Reviewer #1: Yes

Reviewer #2: Yes

6. Review Comments to the Author

Reviewer #1: Authors responded to comments.

Manuscript can be accepted for publication in this revised version.

Reviewer #2: I carefully evaluated the revised version of this manuscript.

Authors have performed the required changes, improving significantly the quality of the paper.

7. PLOS authors have the option to publish the peer review history of their article (what does this mean?). If published, this will include your full peer review and any attached files.

Reviewer #1: **Yes: **Professor Dr. Milan Terzic

Reviewer #2: No

---

## [Editor Report · Acceptance letter]

17 Mar 2023

PONE-D-22-23405R1 

Progestin-primed ovarian stimulation for fertility preservation in women with cancer: a comparative study 

Dear Dr. Filippi:

I'm pleased to inform you that your manuscript has been deemed suitable for publication in PLOS ONE. Congratulations! Your manuscript is now with our production department. 

Kind regards, 

on behalf of

Dr. Gulzhanat Aimagambetova 

Academic Editor

PLOS ONE